# The Next-Generation of Combination Cancer Immunotherapy: Epigenetic Immunomodulators Transmogrify Immune Training to Enhance Immunotherapy

**DOI:** 10.3390/cancers13143596

**Published:** 2021-07-18

**Authors:** Reza Bayat Mokhtari, Manpreet Sambi, Bessi Qorri, Narges Baluch, Neda Ashayeri, Sushil Kumar, Hai-Ling Margaret Cheng, Herman Yeger, Bikul Das, Myron R. Szewczuk

**Affiliations:** 1Department of Biomedical and Molecular Sciences, Queen’s University, Kingston, ON K7L 3N6, Canada; 13ms84@queensu.ca (M.S.); bessi.qorri@queensu.ca (B.Q.); 2Department of Experimental Therapeutics, Thoreau Laboratory for Global Health, M2D2, University of Massachusetts, Lowell, MA 01852, USA; bdas@kavikrishnalab.org; 3Department of Immunology and Allergy, The Hospital for Sick Children, Toronto, ON M5G 0A4, Canada; Narges.baluch@sickkids.ca; 4Division of Hematology & Oncology, Department of Pediatrics, Ali-Asghar Children Hospital, Iran University of Medical Science, Tehran 1449614535, Iran; Ashayeri.n@iums.ac.ir; 5QPS, Holdings LLC, Pencader Corporate Center, 110 Executive Drive, Newark, DE 19702, USA; sushilkumar29@yahoo.ca; 6The Edward S. Rogers Sr. Department of Electrical & Computer Engineering, Institute of Biomedical Engineering, University of Toronto, Toronto, ON M5G 1M1, Canada; hailing.cheng@utoronto.ca; 7Translational Biology & Engineering Program, Ted Rogers Centre for Heart Research, University of Toronto, Toronto, ON M5G 1M1, Canada; 8Program in Developmental and Stem Cell Biology, The Hospital for Sick Children, Toronto, ON M5G 0A4, Canada; hermie@sickkids.ca; 9KaviKrishna Laboratory, Department of Cancer and Stem Cell Biology, GBP, Indian Institute of Technology, Guwahati 781039, India

**Keywords:** immunotherapy, cancer stem cells, oncolytic viral therapy, COVID-19, gut microbiota, drug resistance, combination immunotherapy, CAR T cell therapy, diet, epigenetic therapies, hTERT vaccines, telomerase-targeted immunotherapy

## Abstract

**Simple Summary:**

Drug development and therapeutic approaches for treating cancer have shifted towards incorporating more multimodality approaches that harness the immune system. Despite innovative and notable advances in immunotherapy, challenges associated with variations in patient response rates and efficacies on select tumors minimize the overall effectiveness of these immunotherapy approaches. This review provides an overview of the current immunotherapy options available, followed by epigenetic immunomodulators that may enhance and transmogrify immunotherapy effectiveness. These approaches are positioned to harness trained immunity, improve immune response rates, and increase the efficacy of immunotherapies.

**Abstract:**

Cancer immunotherapy harnesses the immune system by targeting tumor cells that express antigens recognized by immune system cells, thus leading to tumor rejection. These tumor-associated antigens include tumor-specific shared antigens, differentiation antigens, protein products of mutated genes and rearrangements unique to tumor cells, overexpressed tissue-specific antigens, and exogenous viral proteins. However, the development of effective therapeutic approaches has proven difficult, mainly because these tumor antigens are shielded, and cells primarily express self-derived antigens. Despite innovative and notable advances in immunotherapy, challenges associated with variable patient response rates and efficacy on select tumors minimize the overall effectiveness of immunotherapy. Variations observed in response rates to immunotherapy are due to multiple factors, including adaptative resistance, competency, and a diversity of individual immune systems, including cancer stem cells in the tumor microenvironment, composition of the gut microbiota, and broad limitations of current immunotherapeutic approaches. New approaches are positioned to improve the immune response and increase the efficacy of immunotherapies, highlighting the challenges that the current global COVID-19 pandemic places on the present state of immunotherapy.

## 1. Introduction

Cancer accounted for nearly 10 million deaths in 2020 [1]. Over the past few decades, a broad range of therapeutic options has been developed for solid and non-solid tumors, including surgical therapy, chemotherapy, radiotherapy, and targeted therapies. However, despite a small number of actual gains, these therapeutic options have limitations due to the induction of cellular, genetic, and biochemical changes that confer resistance to the treatments [2,3,4]. There is an urgent need to develop highly effective therapeutic approaches to treat cancer that upend tumor cell resistance. For example, neuroblastoma (NB) is a challenging pediatric cancer to treat, given the molecular and pathological heterogeneity [5]. NB is a pediatric developmental cancer that arises in the peripheral sympathetic nervous system’s neuronal ganglia and can present in diverse characteristics depending on the region from which they arise [5,6]. For example, 30% of NB tumors originate from the adrenal medulla, while the other 60% of NB tumors originate from the abdominal paraspinal ganglia [5]. The diversity observed in this malignancy’s clinical manifestation can be attributed to the molecular processes involved in neural crest development, where cancer stem cells have been implicated in this pediatric cancer’s unique pathology [5]. Currently, available treatment options are based on risk categories for NB [7]. Non-high-risk NB patients are reported to have a 90% event-free survival outcome. They are typically diagnosed at <6 months of age with small adrenal cancerous lesions that regress without treatment or are amenable to treatment with chemotherapy and surgical resection [8]. In contrast, high-risk NB patients are diagnosed at >18 months, and treatment approaches include a combination of chemo-, radio- or immunotherapy and surgical resection. However, the high relapse rate can be attributed to the acquisition of drug resistance or clonal selection, with the 5-year survival rate of high-risk NB being <50% [9,10].

Recently, cancer immunotherapy using antibodies that target immune checkpoints has delivered outstanding results. This type of cancer therapy is based on the premise that tumor cells express antigens and harness and modulate the body’s immune system to generate an anti-tumor response [11,12,13]. These tumor-associated antigens may include tumor-specific shared antigens, differentiation antigens, protein products of mutated genes, gene rearrangements unique to tumor cells, and overexpressed tissue-specific antigens and exogenous viral protein expressions. Barth and colleagues recently reviewed the significant advancements in epidermal growth factor receptor (EGFR) specific immunotherapy using armed antibodies [14]. They discuss significant advancements in antibodies modified and used as guiding mechanisms for the specific delivery of readily available chemotherapeutic agents or plants/bacterial toxins, giving rise to antibody-drug conjugates (ADCs) and immunotoxins (ITs), respectively. These armed antibodies provide promising blueprints for further developments in the development of cancer immunotherapy. However, the development of practical therapeutic approaches, including cancer vaccines, has not proven efficacious because these tumor antigens are generally weak immunogens and self-derived antigens.

Another significant problem, especially for immune checkpoint inhibitor (ICI) therapy, is the changing expression of targeted receptors/checkpoints. The challenge is to monitor or map the variety of inhibitory receptors expressed to find the best combinations of drugs aligned with occurring changes in the cancer phenotype, treatment conditions, and timeframes for treatment. For example, the programmed cell death protein 1 (PD-1) antibody and the programmed death-ligand 1 (PD-L1) immunotherapy approach could be rendered ineffective over time due to changes in the expression of the PD-1 receptors. These changes may include modifications to the T cell immunoglobulin domain and mucin domain-3 (TIM-3), which acts as a negative regulator of T helper type 1 (Th1) cell responses, thereby affecting the expression of targetable ligands and receptors. This compensatory mechanism between TIM-3 and PD-1 was observed in lung cancer [15] and melanoma [16]. It appears that this compensatory mechanism, called adaptive resistance, is shared across different types of cancers and leads to reduced responses from effector cells, T cell exhaustion, and decreased T cell survival rates.

PD-1/PD-L1 immunotherapy works well in patients with PD-L1 positive tumors, specifically, when there is no shift in PD-1 expression. Only a small percentage of patients have a positive PD-1/PD-L1 immunotherapy response. Therefore, finding an immune checkpoint for a given cancer with a stable PD-1 expression might be problematic in the clinical setting. Also, there is no universal immunotherapy used in all or most types of cancers. Moreover, the situation can significantly vary from patient to patient, even within the same type of cancer.

The mechanism(s) of these immunotherapy drugs in non-specifically activating T cells can also lead to immune-mediated damage of normal tissues or immune-related adverse events (IRAEs) [17]. IRAEs have been described as affecting nearly every organ system, resulting in colitis, various rashes, pneumonitis, hepatitis, encephalopathy, neuropathy, thyroiditis, cardiac inflammation, and hypophysitis. These IRAEs are some of the wide-ranging adverse effects attributed to ICIs. The report by Bingham III and colleagues highlights cancer immunotherapy-induced rheumatic diseases emerging as new clinical entities [17]. In addition to IRAEs, hyperprogression disease (HPD) has recently been noted in association with the use of ICIs [18,19,20]. HPD is defined as the sudden acceleration of tumor growth kinetics above its baseline growth rate with inherently unstable tumor genomics affecting crucial mechanisms of cell growth [18].

### 1.1. Targets of Immunotherapy

Cancer cell growth and subsequent metastasis are often mediated by immunosuppression and immune evasion. As a result, immunotherapy treatments can be developed to activate the immune system against these malignant cells to prevent cancer progression and improve patient outcomes [21]. There are currently several immunotherapies available as cancer therapies, with more in clinical trials. Immunotherapies can be broadly categorized as immune checkpoint inhibitors, lymphocyte-activating cytokines, chimeric antigen receptor (CAR) T cells, and agonistic antibodies against co-stimulatory receptors, cancer vaccines, oncolytic viruses, and bispecific antibodies. Of these, the most common immunotherapy approaches involve the modulation of immune checkpoints, particularly cytotoxic T lymphocyte-associated molecule-4 (CTLA-4), PD-1, and PD-L1. Other newly emerging targets include inhibitory lymphocyte activation gene (LAG-3), TIM-3, and V-domain Ig suppressor of T cell activation (VISTA), as well as stimulatory inducible co-stimulatory pathways (ICOS, OX40, and 4-1BB) [22].

### 1.2. Limitations of Immunotherapy

Current and developing immunotherapies have shown much promise; however, their clinical use has been limited due to issues regarding the development of resistance, tumor heterogeneity, drug potency, and safety. Additionally, the delivery of and subsequent effectiveness of immunotherapy is dictated by the cancer type, expression of critical biomarkers, and individual immune system competency [23]. These limitations are summarized in Figure 1. Here, we highlight the advantages and limitations of immunotherapy, focusing on improving immunotherapy effectiveness by applying a combinatorial approach. We propose using approaches that target cancer stem cells, oncolytic viral therapy applications, and co-stimulatory methods to enhance T cell response. Given the multifactorial processes involved in cancer progression, monotherapies that target a single pathway require additional therapies to target additional pathways and enhance therapeutic effects against cancer. We have previously reviewed the importance of combining therapies from healthcare, economic, and regulatory approaches and provided an extensive overview of crucial pathways promoting cancer progression and the associated therapeutic agents available to target these pathways [24]. As a follow-up, here, we propose methods to support the function of and perhaps elicit a more robust response from the immune system by introducing epigenetic immunomodulators, dietary supplements, and re-educating gut microbiota to improve the competency of the immune system. Finally, given the global pandemic’s evolving progress, we provide a brief discussion on the emerging evidence of immune responses in cancer patients diagnosed with the novel coronavirus and COVID-19 infection and the challenges and surprising insights provided by this unprecedented situation.

### 1.3. Advantages and Limitations of Cancer Immunotherapy

Our understanding of the immune system and the development of immunomodulation techniques has led to significant cancer treatment improvements. As a result, immunotherapies of many different classes exist. However, despite advancements in immunotherapy research, their clinical use and success have been limited due to efficacy and safety issues [23,25]. As previously noted, only some patients are responsive to immunotherapy, making it difficult to predict patient response [26]. The development of robust immunotherapy resistance further complicates efficacy [27,28]. Detailed reviews of these immunotherapies, which extensively discuss the molecular and cellular mechanisms of action, are out of scope for this review. Briefly, in this review, we highlight some of the most well-studied and clinically used immunotherapies, their associated limitations, and in context, discuss checkpoint inhibitors, cytokines, CAR T cells, and cancer vaccines.

### 1.4. Immune Checkpoint Inhibitors

Some of the most successful anti-cancer therapies are checkpoint inhibitors, with PD-1/PD-L1 blockade and CTLA-4 inhibition being the two most common approaches. Immune checkpoints protect healthy tissues from immune attacks and maintain the appropriate immune response [29]. Concerning their mechanism of action, the PD-1/PD-L1 axis pathway acts in peripheral tissues during the effector phase of the immune response, effectively turning off the immune response after long-term antigen exposure, such as chronic viral infections, to avoid autoimmune damage [30,31,32,33]. Here, the PD-1/PD-L1 activation pathway represents an adaptive immune mechanism of resistance exerted by tumor cells in response to endogenous immune anti-tumor activity. PD-1 is expressed on the surface of immune cells such as CD8^+^ cytotoxic T cells, CD4^+^ helper T cells, B cells, and natural killer cells (NK).

In contrast, PD-L1 is expressed by many different cell types, including tumor cells, lymphocytes, macrophage-lineage cells, and endothelial cells [34,35,36]. When PD-1 binds to PD-L1, PD-L1 acts as a molecular shield for self-tissue that protects cells from being targeted by the immune system [37,38]. PD-L1 overexpressed on the tumor cells binds to PD-1 receptors on the activated T cells, which leads to the inhibition of the cytotoxic T cells. These deactivated T cells remain inhibited in the tumor microenvironment.

Since tumor cells express PD-L1, the biochemical shield formed by PD-1 binding PD-L1 can protect tumor cells from being recognized and targeted by the immune system, essentially putting the “brakes” on the anti-tumor response of the immune system. Briefly, surveillance by T cells leads to recognition of tumor neoantigens as foreign and, upon activation, upregulates PD-1 and secretion of interferon-gamma (IFN-γ). In response to IFN-γ, immune cells and tumor cells express PD-L1, with subsequent PD-L1 binding to PD-1 by “turning off” T cell surveillance. This PD-L1-mediated adaptive immune resistance has been evident in multiple tumor types, including non-small cell lung carcinoma, Merkel cell carcinoma, anal squamous cell carcinoma, breast carcinoma, hepatocellular carcinoma, and melanoma [39,40]. As a result, the PD-1/PD-L1 checkpoint therapeutic blockade using monoclonal antibodies against PD-1 or PD-L1 blocks the PD-1/PD-L1 interaction and activates T cell-mediated cancer cell death. This method has been considered a promising strategy to improve anti-tumor immunity, with durable efficacy in some patients with multiple tumor types [35,41].

The efficiency of the PD-1/PD-L1 pathway to upend the protective shielded blockage against anti-tumor immune responses has been demonstrated in the clinic with anti-PD-L1 ICI blockade in patients with melanoma, non-small cell lung cancer, urothelial carcinoma, Hodgkin lymphoma, head and neck squamous cell carcinoma, renal cell carcinoma (RCC), Merkel cell carcinoma, hepatocellular carcinoma, and gastric carcinomas [35]. The PD-1/PD-L1 immunotherapy works well in patients with PD-L1-positive tumors, and a small percentage of patients have a positive PD-1/PD-L1 immunotherapeutic response.

In contrast, CTLA-4 is a co-inhibitory molecule that regulates the extent of T cell activation. Interactions between CTLA-4 and its ligands, CD80 and CD86, inhibit T cell activity and consequently induce tumor progression [42]. By blocking the interaction between CTLA-4 and its ligands, T cells can remain active to recognize and target tumor cells. However, the exact cellular mechanisms underlying CTLA-4 blockade are now under investigation, with each of the available CTLA-4-targeted antibodies having different mechanisms [43]. For example, some anti-CTLA-4 antibodies have been shown to decrease regulatory T cells and inhibit checkpoint functionality [44,45]. Contardi et al. have reported that CTLA-4 can be constitutively expressed on several tumor cell lines with varying intensities and can trigger apoptosis of CTLA-4-expressing tumor cells after interaction with soluble CD80 (B7.1) or CD86 (B7.2) recombinant ligands [46,47]. The reports provide evidence for apoptosis induction through a caspase-8-dependent mechanism. Interestingly, CTLA-4 expression was detected in osteosarcoma and breast tumor tissues by immunohistochemistry, whereas no or weak CTLA-4 staining was observed in non-malignant breast tissue adjacent to tumors [46].

#### Limitations and Challenges of ICI Therapy

Despite the promising potential of immunotherapy, its clinical use has several limitations, primarily due to the development of acute side effects in organs [48,49,50,51,52]. For example, the mechanism of these immune checkpoint inhibitors (ICIs), non-specifically activating T cells, can also lead to immune-mediated damage of tissue or immune-related adverse events (IRAE). IRAEs with rheumatic phenotypes are increasingly being observed, such as inflammatory arthritis, sicca syndrome, inflammatory myopathy, vasculitis, and lupus nephritis [17]. Furthermore, many patients do not respond to immunotherapy, with factors affecting responsiveness to ICIs attributed to low tumor-infiltrating T cell numbers and deregulation of checkpoints in tumor cells and T cells, the presence of cancer stem cells, and adaptive resistance to ICIs [53,54,55]. Moreover, the tumor microenvironment (TME) heterogeneity results in distinct immunosuppression mechanisms, each of which requires a new treatment approach [56]. Specifically, for anti-PD-1/PD-L1 therapy, patients who are PD-L1-positive often do not respond to therapy attributable to adaptive and constituent patterns of PD-L1 expression. Colocalization of inflammatory response with B7-h1 expression in human melanocytic lesions supports an adaptive resistance mechanism of immune escape. Adaptive PD-L1 expression with ICI was first demonstrated in melanoma and is a dynamic, heterogeneous, IFN-γ driven mechanism in tumor and immune cells [57].

Camelliti et al. have recently reported that, in a fraction of nonresponder patients varying from 4 to 29% based on clinical reports, there was an unexpected increase in tumor growth after ICI administration [20]. This completely unpredictable novel pattern of cancer progression is defined as hyperprogressive disease (HPD). Lau and Leighl [18] reported, based on increasing pre-clinical evidence, that the potential underlying mechanism(s) behind HPD with the inhibition of PD-1 lies in the TME. Here, they identify possible TME cellular populations affected by ICI therapy. For example, regulatory T cells express PD-1 receptors, and under certain conditions, ICI monotherapy with PD-1 inhibition may tip the balance in creating a pro-tumor environment. Other TME cell subpopulations affected by the PD-1 inhibition axis are myeloid-derived suppressor cells (MDSC) [58] and M2 macrophages with immune suppressive activity in the TME [59]. Angelicola et al. [19] reported an elegant review on IFN-y and CD38 in HPD in cancer development. Here, IFN-y is proposed to contribute to hyperprogressive onset by activating the inflammasome pathway, immunosuppressive enzyme indoleamine 2,3-dioxygenase 1 (IDO1) and the activation-induced cell death (AICD) in effector T cells. The role of CD38 in hyperprogressive onset may be associated with the activation of adenosine receptors, hypoxia pathways, and AICD-dependent T cell depletion.

To minimize HPD in immunotherapy in different types of cancers, the combination of cancer immunotherapies may be approached in identifying the presence of hypermutation in tumors. Here, gene profiling to identify the best ICI immunotherapy option(s) would include monoclonal antibodies and tumor-agnostic treatments in combination with oncolytic virus therapy, T cell therapy, and cancer vaccines. Secondly, combination therapy can be used to minimize the intrinsic cytotoxicity and hyper-inflammation side effects of ICI immunotherapy. In combination therapy, other chemotherapeutic agents or repurposed medications can be used with ICI agents to minimize side effects by reducing antibody dosages. The other advantage of this approach is to target multiple cancer survival pathways simultaneously, for example, the TME and cancer-associated factors fibroblasts (CAFs).

To maximize the efficiency of the mechanisms mentioned above, the application of nutrigenomic compounds with proven anti-cancer effects has been suggested [60]. PD-1 inhibition, whether alone or combined with chemotherapy, has little effect on tumor growth, and alternative therapeutic strategies are needed. For example, Mokhatri et al. [60] reported that the phytochemical and bioactive agent sulforaphane (SFN) has nutrigenomic potential in activating the expression of several cellular protective genes via the transcription factor nuclear factor erythroid 2-related factor 2 (Nrf2). Nrf2 is primarily related to mechanisms of endogenous cellular defense and survival. The KEAP1-Nrf2 pathway is considered an important player in tumor progression, where expression of Nrf2-associated antioxidant genes confers protection to the tumor from environmental stress that contributes to chemoresistance and radioresistance. They found that combining acetazolamide (AZ) with SFN reduced tumor cell survival compared to each agent alone, both in vitro and in vivo xenograft tissues. AZ + SFN targeted multiple pathways involved in the cell cycle, serotonin secretion, survival, and growth, highlighting its therapeutic approach. The data suggest that the PI3K/Akt/mTOR pathway is a primary target of AZ + SFN combination therapy. Due to their short half-life, it is proposed to start such compounds before starting immunotherapy, and to continue during treatment. A few good examples of such compounds are sulforaphane, squalene, and polyphenol (−)-epigallocatechin gallate. Nasir et al. [61] reported a comprehensive review on nutrigenomics targeting the epigenetics landscape involved in cancer prevention.

On the other hand, constitutive tumor cell PD-L1 expression refers to a population of tumor cells expressing PD-L1 on their cell surface, independent of an immune infiltrate. Several tumor-intrinsic mechanisms induce constitutive PD-L1 expression due to genetic alterations such as the genomic amplification of 9p24.1 targeting JAK2, PD-L1, and PD-L2 enriched in high-risk triple-negative breast cancer [62,63,64]. It is noteworthy that the frequency of constitutive PD-L1 expression varies by tumor type, with melanoma having infrequent constitutive PD-L1 expression and cutaneous squamous cell carcinoma. A significantly higher proportion of pathology specimens show some degree of constitutive expression [57,65]. Improving response in the PD-L1-positive population may be possible by evaluating whether PD-L1 expression is adaptive or constitutive, and this research is currently underway.

### 1.5. Cytokines

Interferons, interleukins, and granulocyte-macrophage colony-stimulating factor (GM-CSF) are the three major types of cytokines co-administered for immunotherapy [66]. Broadly, injected cytokines improve the growth and activity of immune cells. Interferons are physiologically produced by immune cells in response to microbial pathogens and induce maturation of immune cells such as macrophages, NK cells, lymphocytes, and dendritic cells [67,68,69,70]. Additionally, interferons can also inhibit tumor angiogenesis [66,68,71]. Interleukins stimulate the growth and activity of CD4^+^ helper T cells and CD8^+^ cytotoxic T cells [71,72,73,74]. In contrast, GM-CSF induces T cell homeostasis to improve T cell survival and supports dendritic cell differentiation to express tumor-specific antigens [75].

Although a short half-life limits promising cytokine therapy, cytokine treatment generally requires high-dose bolus injections and, consequently, can induce vascular leakage and cytokine release syndrome (CRS) [66]. Furthermore, cytokine therapy promotes the survival of regulatory T cells and kills stimulated T cells, resulting in an autoimmune attack against normal tissues [76]. For example, IL-2 therapy can cause CRS and vascular leak syndrome, which results in severe fever, hypotension, renal dysfunction, and other potentially lethal side effects [48,66,76]. However, research trends are shifting focus to investigating combination therapy approaches involving two or more cytokines, such as interferons and interleukins, or the combination of cytokines with checkpoint inhibitors or chemotherapies. This combination therapy approach aims to reduce the high dosage of single-agent therapy and its subsequent adverse effects, improving clinical use.

### 1.6. Chimeric Antigen Receptor (CAR) T Cells

The use of CAR T cells as a successful immunotherapy approach has already received FDA approval. For CAR T cell immunotherapy, T cells are isolated from a patient’s blood and genetically engineered to express a chimeric antigen receptor specific for an antigen present on that patient’s cancer cells [52]. Many CAR T cells are first expanded in culture and then re-administered to the same patient. Once in circulation, the CAR T cells recognize the targeted cancer antigen and induce cancer cell death [77]. The first CAR T cell target was CD19, frequently expressed in B cell leukemia and lymphoma. CD19 acts as an adaptor protein to recruit cytoplasmic signaling proteins to the membrane, where it works within the CD19/CD21 complex to decrease the threshold for B cell receptor signaling pathways. Since CD19 expression is confined to the B cell lineage in healthy tissues, B cell aplasia limits this therapy’s adverse side effects. However, this can be overcome with immunoglobulin replacement therapy [78].

The limitations and challenges relating to CAR T cells include severe toxicities, restricted trafficking, infiltration into and activation within tumors, suboptimal persistence in vivo, antigen escape and heterogeneity, and manufacturing issues [79,80]. While many CAR T cell therapy patients are in remission and have prolonged survival, the long-term effects remain unclear and are under current investigation [81]. CAR T cells can cause CRS, CAR T cell-related encephalopathy syndrome (CRES), and neurotoxicity [82,83,84,85]. For example, CRS is common with patients undergoing CAR T therapy in developing multiorgan failure. Also, neurotoxicity/CRES can result in seizures and cerebral edema. Graham et al. [84] propose centers be JACIE (Joint Accreditation Committee-ISCT & EBMT) with accredited bone marrow transplant (BMT) units, alongside hematologists, intensivists, neurologists, cardiologists, and renal specialists.

Under some circumstances, particularly in solid tumors with distinct microenvironments, the injected CAR T cells do not persist, limiting therapy efficacy. However, one of the significant challenges in CAR T cell therapy is the development of CAR T cells’ complex and costly process [86]. Rafiq et al. [79] addressed these crucial CAR T obstacles with a wide range of engineering strategies. For example, optimal molecular design of the CAR T is achievable through examining the many variations of the constituent protein domains. Secondly, the current toxicities associated with CAR T cell therapy can be mitigated, using engineering strategies, to make CAR T cells safer by overcoming on-target and off-tumor toxicities. Thirdly, the efficacy of CAR T therapy can be enhanced with engineering strategies addressing the various challenges relating to hematological and solid malignancies.

In preclinical studies of hematological cancers, Barros et al. [87] develop a mathematical platform to enable in silico experiments to investigate the interplay between tumor cells, effector, and memory CAR-T cells in immunodeficient mouse models. They found that CAR T therapy effectiveness mostly depends on the differentiation of effector to memory CAR T cells, CAR T cytotoxic capacity, tumor growth rate, and tumor-induced immunosuppression. Interestingly, using the HDLM-2 tumor model with low tumor proliferation and a less aggressive tumor, CAR T cell therapy was effective on tumor elimination and enhancing memory CAR T cells. The model captured tumor elimination after CAR T immunotherapy with new tumor challenges due to memory CAR T cells’ long-term protection. In contrast, the RAJI model, with its high proliferation and escape from CAR T cell immunotherapy, identified the effect of indoleamine-2,3-dioxygenase 1 enzyme (IDO1) overexpression by the RAJI cells, impacting CAR T cell immunotherapy and their combination with an IDO inhibitor. Importantly, it was noted that the CAR T cell dose determination for a given tumor burden is a critical factor in immunotherapy success [87].

### 1.7. Cancer Vaccines

Cancer vaccines can be prophylactic, preventing cancer, or therapeutically eradicate pre-existing cancer. There are generally two components to a cancer vaccine—a specific tumor antigen and an adjuvant capable of generating an immune response [88]. Adjuvants stimulate dendritic cells’ maturation, presenting tumor antigens in the vaccine on the major histocompatibility complex (MHC) surface to induce an anti-cancer T cell response. Gardasil^®^ (Merck & Co., Inc., d.b.a. Merck Sharp & Dohme (MSD), Kenilworth, NJ, USA) and Cervarix^®^ (GlaxoSmithKline plc (GSK), London, UK) are prophylactic vaccines against the human papillomavirus (HPV), while Sipuleucel-T is a therapeutic vaccine for metastatic prostate cancer [89]. Sipuleucel-T preferentially targets the prostate acid phosphatase (PAP) antigen, expressed on 95% of prostate tumors, and is delivered with GM-CSF to activate antigen-presenting cells (APCs) [25]. Following treatment, host T cells target the PAP. Although this approach has demonstrated clinical efficacy, one of the primary limitations of cancer vaccines is identifying tumor antigens that will elicit a robust anti-tumor response. Additionally, this approach requires a competent immune system, and, as many cancer patients have a compromised immune system, cancer vaccines may not be as effective.

### 1.8. Methods to Improve the Effectiveness of Immunotherapy

Despite the variations observed in patient response rates, robust research is underway to investigate the underlying factors contributing to these disparities and offer solutions and new therapeutic approaches to overcoming immunotherapeutic limitations. Specifically, targeting cancer stem cells, incorporating oncolytic viral therapy, and introducing dietary supplements to enhance the immune system’s function are methods that have been recently proposed and will be discussed in detail and presented in Figure 2.

### 1.9. Targeting Cancer Stem Cells to Enhance the Effectiveness of Immunotherapy

Cancer stem cells (CSC) have been identified as a subpopulation of cells that play a major role in cancer initiation due to their tumorigenic [90,91,92,93] self-renewal [94,95,96,97,98], slow-cycling [99], and multilineage differentiation potential. As a result, CSCs represent a promising target for preventing cancer relapse and improving patient survival rates [91,96,100,101]. Although undifferentiated CSCs comprise only a minority of the tumor cell population, their unique differentiation capability can generate all of the differentiated progeny of tumor cells [102,103]. CSCs have been detected in solid and non-solid tumors, including leukemia, brain, lung, head and neck, colon, breast, and pancreatic tumors [104,105].

CSCs require a suitable environment and are found within the CSC niche, a distinct environment that supports the balance between self-renewal, activation, and differentiation [106,107,108,109]. These niches promote and regulate stemness, proliferation, and resistance to apoptosis [110]. The CSC niche can protect cells from anti-tumor therapies [111,112,113]. Hypoxic conditions induce CSC proliferation and repeated self-renewal, with the self-renewal capacity of CSCs declining with increasing oxygen concentrations [114]. Hypoxia also alters the microenvironment, with malignant cells in hypoxic conditions overexpressing carbonic anhydrase 9 (CA9), which catalyzes the reversible hydration of carbon dioxide to yield bicarbonate and lose protons. The increased concentration of protons surrounding the tumor cells lowers the pH in their environment and results in stromal acidosis, which allows cancer cells to escape the tumor of origin and migrate to other organs in the body [115]. As a result, CSCs generally do not respond to chemotherapy or radiation. Thus, while anti-cancer drugs may successfully target differentiated cancer cells, they will not be efficacious against CSCs [116]. Consequently, many patients experience pharmacological resistance in response to conventional therapies, with the potential for relapse after a disease-free period or metastatic dissemination [91,97].

Lymphanogenous and hematogenous spreading of cancer cells contributes to metastatic spread associated with poor clinical outcomes and is a significant cancer treatment complication. Different CSC markers have been associated with contributing to metastatic potential and progression [117]. For example, the aldehyde dehydrogenase (ALDH1) expression level was significantly higher in gastric cancer lymph node metastases than in the primary tumor [118]. ALDH1 has also been reported to be the most significant cell marker in potential neuroendocrine CSCs (N-CSCs) [119]. In another study on colorectal cancer, single cell-derived progenies from CD133^+^ colorectal cancer monoblast cells were shown to have more tumorigenic potential and displayed heterogeneity in metastatic and invasive potential [120].

It has been reported that CSCs can undergo epithelial-to-mesenchymal transition (EMT) to convert into migrating CSCs that have the potential to metastasize and form colonies [121]. Transdifferentiation of CSCs can activate angiogenic pathways that promote tumor propagation in glioblastoma. It was believed that CSCs in glioma were from the neural lineage, but it has been shown that there exist glioma CSC subpopulations from both the neural and mesenchymal lineages [122]. Stromal matrix metalloproteinase-9 can promote pericyte recruitment in neuroblastoma, causing vascular colonization and angiogenesis [123]. Moreover, CSCs have been found to express neuronal phenotypic markers such as gamma-aminobutyric acid (GABA). This molecular mimicry has only been observed in brain tumors or other favorable microenvironments that contain neurons near a tumor [124]. Breast tissue is an example of a tumor-favorable microenvironment. Breast tumors often metastasize to the brain, and these metastases express high concentrations of GABA receptors, GABA transporters and transaminases, and brain-specific proteins such as parvalbumin [124].

Additionally, we have recently reported on a fraction of stem cells that exhibit altruistic behavior and could provide additional insights into the activity of CSCs [125,126]. When exposed to extreme hypoxia/oxidative stress, a few CSCs from the heterogeneous population expand and exhibit reprogramming to a higher state of stemness. This reprogramming state, or enhanced stemness reprogramming, exhibits an altruistic behavior. In brief, this stem cell altruism is a defense mechanism that is engaged to protect surrounding stem cells that are genetically “weak”. Altruistic stem cells can downregulate p53 and produce proteins that support weaker counterparts’ fitness or undergo apoptosis or differentiation to support tissue regeneration [127]. CSCs can engage similar mechanisms. Thus, identifying and eradicating CSCs represents a challenging but promising target that could destroy cancer in its entirety. The elimination of the CSC population would prevent metastasis, decreasing the possibility of tumor regeneration [97,128]. However, their instability and complex biology limit CSCs as a target for immunotherapy [129,130]. The unique biology and presence of CSCs contribute to the variations observed in immunotherapy responses, particularly concerning resistance to ICIs [131]. It is postulated that one of the mechanisms of resistance to immunotherapy is the activity of genes associated with EMT, angiogenesis, and stemness of CSCs that prevent T cell recognition, leading to the failure of immunotherapies [132]. As detailed, CSCs have been characterized as expressing high levels of anti-apoptotic proteins, such as B-cell lymphoma 2 (Bcl-2) and survivin, which protect them from apoptosis-inducing immune effector T and NK cells [133]. This is accomplished through ICAM-1-mediated decrease of PTEN, which in turn leads to the activation of the PI3K/AKT pathway and is proposed to allow CSCs to evade CTL-mediated clearing of cancer cells [134]. Additionally, CSCs were also characterized by a lack of MHC 1 ligands, allowing these cells to evade clearance by NK cells [135].

CSCs are undifferentiated cells that can undergo EMT to adapt to their environment. This undifferentiated state has been shown to affect immunotherapy’s effectiveness, where only differentiated melanoma cells expressed a tumor-associated antigen that T cells could recognize [136]. Furthermore, CSCs have been documented to evade immunotherapy’s therapeutic effects with high PD-L1 expression and promote a highly immune-suppressive microenvironment in a manner more robust than typical tumor cells [133,137]. For example, breast CSCs and glioblastoma stem cells have been characterized as secreting higher levels of TGF-β. In contrast, colon cancer stem cells secrete high levels of IL-4 [133,138,139]. Collectively, CSC’s ability to evade immune-mediated clearing of cancer cells represents a unique challenge that likely contributes to the varied response rates observed at the clinical level. Therefore, strategies for and modifications to current immunotherapy options offer a solution to targeting CSCs and improving immunotherapeutic response rates.

One of the most common solutions to targeting CSCs is rooted in T cells’ antigen-specific targeting. Several antigens are commonly expressed on CSCs that are attractive targets, including ALDH, CD133, and EpCAM, to name a few [140]. To test this, ALDH^hi^ lung CSCs were isolated and were exposed to dendritic cells (DCs), followed by co-culturing with activated CD8^+^ T cells [141]. This study demonstrated increased survival and decreased tumor volume following treatment with ALDH-targeting T cells. As outlined above, CAR T cell immunotherapy involves priming T cells to CSC-specific antigens or genetically engineering T cells to express chimeric antigen receptors (i.e., CAR T cell immunotherapy) [140]. This approach offers a more controlled method to genetically manipulate T cells so that they can recognize tumor-associated antigens (TAAs) in an MHC-unrestricted manner [142].

## 2. Training Immunity with Epigenetic Immunomodulators

The concept of trained immunity or innate immune memory predicts a long-term functional reprogramming of innate immune cells. This reprogramming is activated by exogenous or endogenous insults, leading to an altered response towards a secondary activation after returning to a non-activated state. Trained immunity is defined by specific characteristics, as recently reported by Netea et al. [143]. Firstly, the cellular components mainly involve myeloid cells, natural killer (NK) cells, and innate lymphoid cells (ILCs) together with specific germline-encoded recognition and effector molecules, such as pattern recognition receptors and cytokines, that are different from those involved in classical immunological memory. Secondly, the increased responsiveness to secondary stimuli during trained immunity is not specific for a particular pathogen. Instead, it is mediated through signals involving transcription factors and epigenetic reprogramming and rewiring, leading to changes in cellular mechanisms not involving permanent genetic changes, such as mutations and recombination. Thirdly, trained immunity uses altered functional states of innate immune cells that last for weeks to months, rather than years, after the initial stimulus. Fourthly, the potential roles of the environmental epigenetic changes that involve developing a roadmap of epigenetic networking, such as dietary components on epigenetic imprinting and restoring DNA methylation patterns laid down during embryonic development, to establish youthful cell type and function [144], exploring alternative nutrition-based therapeutic approaches, and developing tools for a personalized diet to improve health and increase life expectancy [145].

### 2.1. Oncolytic Viral Immune Training

Oncolytic viral therapy represents a subcategory of immunotherapy that involves using a virus to preferentially infect cancer cells to elicit an immune response that clears cancer without harming healthy host cells [146]. Several factors need to be considered when identifying a viable candidate virus to treat cancer [147]. The candidate virus must penetrate the cancer cell membrane, inhibit host protein synthesis, and sufficiently self-replicate to cause cancer cell lysis [148]. These include the necessary steps to release tumor antigens and elicit a robust immune response.

In most studies on virotherapy, the virus infects cancer cells and self-replicates in the tumor. The inactive antiviral defense response in malignant cells is reactivated to attract immune cells [149]. There is currently only one approved oncolytic viral therapy, called T-VEC (Imlygic^TM^), which treats a small subset of patients presenting with non-resectable metastatic melanoma [147]. T-VEC is a recombinant human herpes simplex virus type 1 (HSV1) engineered to encode two copies of the GM-CSF gene. It has been administered in conjunction with checkpoint inhibitors to enhance its anti-tumor effects and elicit a robust immune response [147].

Additionally, Rigvir and Oncorine (H101) have been commercially approved to treat cancer in parts of Europe and China, respectively [147]. Rigvir is an unmodified enteric cytopathogenic human orphan type-7 (ECHO-7) picornavirus used to treat melanoma in Latvia, Georgia, and Armenia [150]. The available literature on the efficacy of Rigvir was reported to be effective in treating low-grade melanomas, but data on high-grade melanomas remains inconclusive. Additional data is needed to explore the underlying mechanism of action of Rigvir [147]. Oncorine (H101), on the other hand, is a modified serotype 5 adenoviral vector, capable of replicating in p53-deficient tumors. Additional iterations of this oncolytic virus allow it to infect and replicate in typical p53 tumors [151]. When administered in conjunction with chemotherapy, clinical data demonstrated a 78.8% response rate versus chemotherapy alone (39.6%) [152].

Although oncolytic viral therapy is currently in a state of infancy, several candidate viruses for virotherapy are currently under investigation, including the measles virus [153], Newcastle disease virus [154], and adenoviruses [155]. The rationale for using both virotherapy and checkpoint inhibition follows the premise that the oncolytic viral infection of cancer cells would create a pro-immunogenic TME, allowing the infiltration of immune cells, where “cold” tumors become “hot” tumors, and thereby enhancing the efficacy of checkpoint inhibition [156]. Several clinical trials are currently underway assessing the efficacy of combination virotherapy and checkpoint inhibition, and results are forthcoming [156].

### 2.2. Zika Viral Immune Training 

Recently, the Zika virus (ZIKV) has become a candidate of interest for treating brain cancers. It is one of a few viruses that can cross the blood-brain barrier and has been detected in cerebrospinal fluid [157]. Characterization studies have reported that the ZIKV uses proteins, including receptor tyrosine kinase AXL and its ligand Gas6, to enter and infect human glial cells [158] and neural stem cells [159]. Following the entry into cells, the ZIKV is proposed to use Musashi (MSI-1), the RNA-binding protein, to self-replicate in the host [160]. Additionally, the MSI-1 protein has been documented to increase proliferation by modulating cyclin-dependent kinase activity and is enriched in neural stem cells [160]. These proteins are significant because they are overexpressed in cancers, specifically in glioblastoma multiforme (GBM) [149]. Initially promising data have revealed that this virotherapy is efficacious because the virus can target glioblastoma stem cells [161]. In this seminal report, the Zika virus-induced apoptosis in cancerous neural cells increased survival rates in mouse models of glioblastoma. However, this study also reported that healthy neural cells were also infected. Thus, further modifications to the ZIKV need to be assessed to reduce its pathogenicity to normal neural cells. As of June 2020, there were 95 clinical trials that were identified, with 50% in phase I, 6.2% in phase I/II, 11% in phase II, and, finally, only 2% in phase III clinical trials [162]. Of these 95 trials identified by Macedo et al., the most common cancers targeted included melanoma, head and neck cancer, breast and other gynecological cancers, and sarcomas [162]. For example, one trial (NCT03206073) is an ongoing phase I/II study assessing Pexa-Vec (JX-594), a vaccinia virus engineered to express GM-CSF beta-galactosidase in combination with CTLA-4 immunotherapy in treating metastatic colorectal cancer. This ongoing trial has not reported any results as this trial is slated to be completed by December 2021.

### 2.3. COVID-19 Viral Immune Training in Convalescing Patients

Given the global impact of the current COVID-19 pandemic, it is necessary to consider the challenges of infection with the novel coronavirus (SARS-CoV2), which has not been applied as an oncolytic viral therapy, which may negatively impact the current state of immunotherapy. Patients with cancer have a compromised immune system both because of the malignancy itself and treatment options that affect immunity. Therefore, understanding the implications of a cancer patient’s susceptibility to the COVID-19 virus is clinically relevant due to the nature of this virus’ global transmissibility. An analysis of patients from China revealed that 1% of patients diagnosed with COVID-19 had a history of cancer and, perhaps not unexpectedly, experienced severe COVID-9 outcomes [163]. Although additional data will require a complete understanding of cancer patients’ susceptibility to the COVID-19 virus, the altered immune responses suggest a poor clinical outcome. COVID-19 is a member of the severe acute respiratory syndrome family of RNA viruses that commonly affects the respiratory tract and is currently spreading globally. Over 95 million cases have been reported worldwide since the pandemic began in December 2019, and these have accounted for over 2 million deaths. Given the contagious nature of this virus and its subsequent effects on the immune system, it is vital to consider the new challenges this virus presents for cancer patients [164]. Therefore, a brief overview of the disease and the associated immune response will also be provided, followed by a discussion on the critical considerations concerning immunotherapy responsiveness of cancer patients.

Coronaviruses are composed of a lipid bilayer envelope that surrounds an RNA genome in addition to proteins that are important for its viral activity, including the spike (S) protein, which is required to invade, attach, and enter human cells [165]. The S-protein has an S1 domain, which binds and interacts with the angiotensin-converting enzyme 2 (ACE-2) located on human cells [165]. The S2 domain, on the other hand, is required for membrane fusion between the host cell and the virus. Once the host cells are infected, several immune responses are reportedly engaged to clear the pathogen. Infected lung epithelial cells produce IL-8 to attract neutrophils and macrophages, after which the adaptive immune system triggers infiltration by T and B cells [166]. Following this immune response, pathogen-associated molecular patterns (PAMPs) further amplify the innate immune response by using ssRNA virus particles to trigger Toll-like receptor 7 (TLR7) to present the pathogen to dendritic cells and macrophages, which then activate the pro-inflammatory cytokine signaling pathways [167]. Creating an inflammatory environment is intended to expedite the infection’s clearance; however, COVID-19 has revealed unique immune responses and outcomes that are not typical of previous coronaviruses.

It is essential to consider the added challenge associated with immunotherapy treatment options for cancer patients with COVID-19. As thoroughly detailed, immunotherapies exploit the host’s immune response to exert therapeutic effects. Given that the immune system would be engaged in exerting a robust immune response to clear cancer cells, the possibility of a dampened response to a subsequent or simultaneous COVID-19 infection may compromise the effectiveness of the immunotherapy. For example, T cell exhaustion has been documented in patients receiving immunotherapy. It is also typical of chronic viral infections, where the effector function of T cells is reduced and has a lower proliferative potential. For example, Diao et al. [168] found a total of 19/20 COVID-19 patients that demonstrated a decrease in total T cells and CD4^+^ T cells, whereas all patients displayed a decrease in CD8^+^ T cells.

Interestingly, the T cells from these patients had significantly higher levels of the exhausted PD-1 marker. Also, there was an increased expression of PD-1 and TIM-3 markers on the T cells, as patients progressed from early signs or symptoms of illness to overtly symptomatic stages. This could ultimately render immunotherapy an ineffective option for cancer patients who would otherwise be good candidates for this treatment option.

Given the highly inflammatory environment produced during an immune response to COVID-19 infection, an additional challenge to consider would be the effects of these cytokines and proteins on CSCs. The relationship between the activation and proliferation of CSCs and an inflammatory microenvironment has been a highly debated topic and continues to be explored [169]. In certain cancers, such as breast cancer, inflammation has been associated with cancer development and subsequent progression [170]. Given that CSCs have been reported to facilitate tumor growth and metastasis and inflammation regulating their activity, the challenges associated with cancer patients with COVID-19 introduce an added clinical obstacle. There are currently no therapies that specifically target this subpopulation, and given their plastic nature, they can respond to local and systemic environmental cues.

Collectively, the obstacles presented by cancer patients contracting COVID-19 require a unique treatment approach that considers the natural immune response to infection, creating both an inflammatory environment and the subsequent effects of these obstacles on immunotherapy treatment. As outlined in this review, adopting a combination approach to supporting the immune system, so that a robust response can eradicate both cancer cells and a viral load on the host, is essential to consider when managing cancer patients. Although the evolving pandemic has led to several unanswered questions from an oncological perspective, unique cases have also provided insights into the immune response’s power. A recent case report on a patient diagnosed with non-Hodgkin’s lymphoma who subsequently contracted COVID-19 is one example [171]. This patient reportedly experienced a complete remission due to an anti-tumor immune response triggered by COVID-19. Although this was an unexpected result, a healthy immune system’s underlying importance cannot be overlooked.

### 2.4. Bacterial Immune Training

A new class of immunotherapy has recently incorporated the use of bacteria as a potential treatment option and has been gaining attention for its immune-regulating mechanism of action. The bacillus Calmette–Guerin (BCG) vaccine, initially used to protect against tuberculosis, is the current gold standard immunotherapy option used to treat bladder cancer [172]. Treatment with this vaccine leads to an innate immune response that leads to anti-tumor responses and prevents disease recurrence. Most recently, BCG applications have been considered to target CSCs [173]. As outlined, Das et al. were the first to report that CSCs reside in the hypoxic niche, which has been implicated in the enrichment of CSCs. This hypoxic population was then used as a marker to identify CSCs [174]. However, therapeutic targeting of this cell subpopulation is an area of intense research focus. Therefore, BCG’s possible application in targeting CSCs is an exciting frontier that holds promise. Our group has reported promising results using mesenchymal stem cells infected with BCG to target CSCs in the hypoxic tumor microenvironment. This approach not only has the potential to overcome drug resistance conferred by CSCs, but has been demonstrated to transfer BCG to cancer cells [175]. Though in the early stages, this novel application provides an opportunity to overcome the limitations of other therapies.

## 3. Immunomodulators Targeting Epigenetic Regulators

### 3.1. CpG Nucleotides: Combination Options to Increase the Effectiveness of Checkpoint Inhibitors

Limitations associated with cancer vaccines and checkpoint blockades contribute to the variable and incomplete patient responses following monotherapy. For example, cancer vaccines require a specific tumor antigen that the patients’ immune system can target once antibodies to this antigen are produced [176]. The success of this approach hinges on the presence of the antigen in the patient’s tumor. Similarly, the success of PD-1/PD-L1 and CTLA-4 checkpoint inhibitors require a competent immune system and tumor infiltration by T cells so that they may be reactivated to continue their anti-tumor response. Therefore, many of the varied clinical responses observed with this therapeutic approach include a lack of immune system competency and diversity, cold tumors, and T cell exhaustion, as previously discussed. Here, we propose applying a combination and personalized approach that offers a method to overcome some of these limitations.

The therapeutic efficacy of checkpoint inhibitors is derived from releasing the “brakes” on T cells in the TME to facilitate T cell-mediated anti-tumor responses. Unfortunately, releasing the “molecular brakes” is one of two essential steps required to launch a robust immune response. The second step allows the immune system to elicit a more potent immune-mediated clearance of tumor cells [177]. This method leads to localized anti-tumor effects. However, it can clear cancerous lesions at distant sites because of its systemic response [178,179]. The rationale for this approach exploits the interactions between innate and adaptive immune responses to enhance T cells’ activity in the local tumor distant sites.

Briefly, intratumoral injection of the CpG nucleotide sequence leads to a localized response combined with monoclonal antibodies that modulate local and system T cell responses [179]. The CpG nucleotide sequences are a ligand for Toll-like receptor-9 (TLR-9), expressed on immune cells including dendritic cells (DC), macrophages, NK cells, and other antigen-presenting cells, and initiate a cellular response when activated by this ligand [180]. Injection with CpG at the tumor site has demonstrated efficacy in increasing DC infiltration and has been shown to inhibit tumor cell proliferation [181]. Vaccination with CpGs offers a personalized approach that essentially overcomes the limitations associated with cancer vaccines. With this approach, a patient’s immune system would present tumor-associated antigens unique to their tumor. Thus, they would theoretically be immunized against multiple antigens rather than a single antigen typical of commercially available vaccines [179]. However, as promising as this approach appears to be, supplementing this treatment regimen with monoclonal antibodies that can modulate T cell responses provides a method to launch a localized response while ablating cancerous lesions at distant sites. For example, modulating T cell responses can increase T cells’ anti-tumor activity, decrease Tregs that suppress the immune response, and activate T cells whose action has been inhibited by the immunosuppressive microenvironment [179].

Although several monoclonal antibodies have been proposed and tested, anti-OX40, a tumor necrosis factor receptor family member, has reportedly been the most effective in eliciting a robust immune response [177,178,179]. Following the interaction between the T cell receptor (TCR) and MHC, co-stimulatory receptors activate T cells to proliferate and survive [182]. One such co-stimulatory molecule is OX40, expressed on activated T cells inducing T cell proliferation and expansion [177]. The upregulation of OX40 has significant implications for several reasons. For example, Lane [183] demonstrated that CD28-dependent OX40 signals from DCs profoundly involve the coordination of the selection, migration, and cytokine differentiation of both Th1 and Th2 CD4+ cells, which induce a rapid expansion of antigen-specific cells in the T cell zone, accompanied by their selective migration to B cell germinal center follicles. A significant portion of CD4+ T cells in the tumor microenvironment express OX40 due to the recognition of tumor antigens, which presents an attractive therapeutic approach for anti-OX40 treatment [184]. Reports have suggested that OX40 may modulate Tregs’ suppressive and expansion function [185,186]. Collectively, the ability of OX40 agonists to modulate adaptive immune response led to investigations into combining anti-OX40 with CpG vaccination to enhance immunotherapy [178].

In a seminal report by Sagiv-Barfi et al. [178], following intratumoral injections with CpG nucleotides and subsequent treatment with anti-Ox40, not only was a decrease in tumor volume at the injection observed, but untreated tumors at distant sites also demonstrated decreases. These findings are supported by another report by Houot and Levy [179], in which a similar treatment regimen for lymphoma xenograft models was administered and a decrease in tumor volume without the need for chemotherapy was reported. Although anti-OX40 monotherapy is currently being tested in clinical trials, the combination approach, with CpG vaccination, requires further in vivo study before assessing this treatment regimen’s efficacy. This dual therapeutic approach collectively provides a personalized method to overcome the limitations of cancer vaccines and checkpoint inhibitors.

### 3.2. Restoring Epigenetic Reprogramming with Immunotherapy to Improve Therapeutic Responsiveness

It is well established that epigenetic mechanisms play a critical role in the normal physiological process, such as B cell development and forming antibodies through VDJ recombination to generate a diverse immune system [187]. However, cancer cells also employ several strategies to evade the host immune response, including epigenetic mechanisms such as hypermethylation of CpG islands [23]. Although the premise of conventional immunotherapies is to reverse the dampened immune response generated through typical immune evasion mechanisms (i.e., checkpoint inhibitors), current immunotherapies do not actively target epigenetic mechanisms that contribute to immune evasion by cancer cells.

Epigenetic mechanisms play a significant role in the TME and contribute to immune cell evasion [188]. The concept of epigenetic regulation in the TME has been extensively reviewed elsewhere [187]. In brief, cancer cells utilize mechanisms, including DNA methylation and histone post-translational modification, to modify the expression of identifiable tumor antigens, thereby evading immunosurveillance. As a result, agents that target critical epigenetic mechanisms have been an emerging area of interest in recent years [189]. Moreover, coupling epigenetic therapies with immunotherapies has significant implications in improving overall patient responsiveness and is in the early stages of exploration in clinical trials [190].

Current epigenetic therapies include DNA methyltransferase (DNMTs) and histone deacetylase (HDACs) inhibitors [190]. The use of these agents has shown promise in upregulating tumor-associated antigens (TAAs), which are typically downregulated by tumors to avoid immune surveillance [190,191]. For example, DNMT inhibitors can cause demethylation of sites that downregulate the expression of antigens, such as the cancer-testis antigens (CTAs), leading to re-expression in tumor cells and the promotion of immune cell recognition [192,193]. HDAC inhibitors cause a similar outcome. CTA expression can be induced and lead to immune response [194]. HDAC inhibitors are currently approved to treat cutaneous and peripheral T cell lymphoma [195]. In contrast, DNMTs are currently approved for hematologic cancers [196].

Alternatively, epigenetic regulators can also modulate host immune responses and develop components of the immune system [190]. For example, treatment with HDAC inhibitors has shown to directly affect the host immune cell response by increasing the antitumor activity of T and B cells [197]. Therefore, the use of epigenetic therapies to reverse the dampened expression of key tumor antigens combined with immunotherapies could lead to a robust immune response by overcoming epigenetic tumor evasion and enhancing antitumor activity. This strategy merits further exploration to confirm the clinical efficacy of this approach and continues to be an exciting area of research.

### 3.3. Exercise-Induced Epigenetic Modifications in Genes

Regular physical activity can help to improve overall health. Increasing evidence supports the premise that exercise can result in, or can be mediated by, epigenetic modifications [198]. Environmental factors, including diet [199] and physical exercise [198], have been reported to alter the epigenome. Barrón-Cabrera et al. [200] reported an excellent systemic review on the epigenetic modifications resulting from exercise interventions. Interestingly, the report describes exercise-induced modifications in genes related to (a) insulin resistance and type-2 diabetes, (b) obesity, (c) inflammation, and (d) cardiovascular disease and blood lipid alterations. These studies indicate that exercise interventions can alter the epigenome and that these outcomes could be related to specific metabolic pathways. However, there are several limitations in the studies reported to date, including small sample size, heterogeneous populations, different exercise interventions, the exercise tests available, and the different epigenetic modifications measured in different tissues. Due to the differences and complexity of the existing literature, a specific recommendation about the type, intensity, or duration of exercise that could be beneficial for different subsets of the population such as healthy, diseased, and trained individuals is currently not possible.

Interestingly, exercise and immunotherapy can have a synergistic effect. Reports have indicated that exercise can lead to mobilization of the immune cells, resulting in their redistribution to different body compartments [200,201]. In preclinical models, exercise has led to immunological changes in the tumor microenvironment [201]. The report suggests that acute exercise is an essential immune system adjuvant to stimulate the ongoing exchange of leukocytes between circulation and tissues. In contrast, exhaustive exercise and high-intensity training seen in athletics have an adverse increased risk of illness [202]. It is noteworthy that studies with athletics in marathon exercise were reported to be associated with an inflammatory profile and increased risk of injury. The in silico analysis supported an association between the observed soluble inflammatory mediators and circulating-inflammatory miRNA profiles and the pathways of cancer, immune system disorders, and inflammation process [203]. These observations suggest an essential difference between acute-exercise-induced stress and activation of the immune system versus chronic exercise-induced stress.

### 3.4. Targeting Cancer Stem Cells Through Telomerase-Targeted Immunotherapy

Another relatively recent therapeutic area of exploration targets telomerase, an enzyme critical to telomeres and cancer cell replication production [204]. Given the vital role played by telomerase, recent efforts have focused on the development of inhibitors and immunotherapeutic approaches that can target this enzyme [205]. The recent discovery that a mutation in the human telomerase reverse transcriptase (hTERT) gene was associated with increased telomerase activity and tumor progression has been a driver of research in therapeutically targeting this mutation [206]. For this review, we will limit our discussion to immunotherapeutic strategies to target this mutation.

The premise of immunotherapeutic targeting of hTERT is that this protein is overexpressed in 85% of tumors, is highly expressed in CSCs, and presents as a TAA that could be targeted by immune cells [207]. This is because regular hTERT expression is limited in many cells, including testicular cells and hematopoietic stem cells [208]. Studies have shown that short and long peptides from hTERT can complex with MHC class I and II molecules and launch an immune response, and therefore has implications as a global TAA that can exclusively target cancer cells, including the ever-elusive CSC population [209].

Furthermore, cancer vaccines that target hTERT peptides are also currently in development, including GV1001, GX301, UV1, and Vx-001.183. These vaccines have broad applications in lung cancer, pancreatic cancer, and melanoma [210]. The efficacy of GV1001 has been shown in early animal studies [211] and clinical trials [212]. For example, a study on pancreatic cancer xenograft models treated with conventional chemotherapeutics and GV1001 demonstrated a significant reduction in tumors, as evidenced by increased tumor cell death [211]. Additionally, a phase I/II clinical trial investigating the clinical response of GV1001 in patients with lung cancer demonstrated promising results, where several patients experienced an immune response, with one patient exhibiting a complete response [212]. GX301, UV1, and Vx-1001 are all currently in the early stages of study. They have been used to assess the efficacy of multi-peptide vaccines [213], combinations of vaccines with cytokines [214], and as a single vaccination study [215], respectively. Finally, CAR T cell therapies have also been considered as a viable immunotherapeutic approach to target hTERT; however, this is an area that is currently in a state of infancy and continues to evolve [205]. Collectively, these advances in therapeutically targeting telomerase with immunotherapeutic strategies exemplify a promising area of exploration that will have a significant clinical impact as investigations continue.

## 4. Harnessing the Gut Microbiota to Enhance the Immune Response

### 4.1. Diet and Nutrient Influence Epigenetic Modifications

Zhang and Kutateladze [216] discuss three studies, recently published in Nature Communications [217,218,219], that have explored how diet or compounds found in food can alter gene expression programs through epigenetic mechanisms. The question is: how might food consumption influence epigenetic modifications that would impact health? One possibility is directly affecting the enzymes responsible for ‘writing’ or ‘erasing’ the epigenetic modifications. For example, the plasticity of cancer cells involves their unique capacity to react and adapt to prominent environmental cues, and their ability to retain stable phenotypic changes after exposure to stimuli. Nonetheless, many of these signals can be simultaneously present in inflamed tissues. It is not clear how signaling is integrated during cell fate decisions or whether cells can reverse their phenotypes in response to conflicting polarization cues [220]. Rodriguez et al. [220] have reviewed and proposed that epigenetic mechanisms, such as DNA methylation and histone marking, are critical during the differentiation and activation of immune cells, indicating that chromatin can act as an integration node during myeloid cell polarization, under physiological conditions, as well as in inflamed/damaged tissue sites. It is conceivable that pro-inflammatory cytokine production, polarization, tolerance, and innate immune memory might be targeted at the epigenetic level to treat various conditions, offering new putative therapeutic options.

One of the most critical determinants of a successful response to immunotherapy is the immune system’s functionality and competency. For example, a greater diversity of HLA class I molecules has been shown to coincide with more excellent responsiveness to immune checkpoint blockade through anti-PD-1 or anti-CTLA-4 therapies [221]. Therefore, supporting the immune system and improving the host’s immune system through epigenetic dietary supplements could enhance immunotherapies’ overall effectiveness. It is well documented that a healthy diet can promote overall good health. For example, Mediterranean, Japanese, and vegetarian diets, microbiota-regulating diets, and ketogenic diets have been associated with a lower risk of developing several cancers [222,223], and there is also an association of the Mediterranean diet with SARS-COV-2 infection [224]. It is proposed that one role of nutrition is improving responses to cancer therapies. Thus, understanding the pathways by which nutrition influences the immune response can shed light on innovative holistic immunotherapy treatments.

The presence or absence of metabolic factors can influence immune cell activity within the TME. For example, supplements such as omega-3 and polyphenols are beneficial concerning response to immunotherapy alongside the benefits in dietary and lifestyle habits [223]. More broadly, metabolic status can alter body weight, influencing immune status. Although the mechanism of action has not been conclusively established, vitamins A, B, C, D, and E have been implicated in NK cells’ stimulatory effects [225]. For example, in anemic Japanese patients, vitamin B12 was implicated in improving the cytotoxic activity of NK cells [138] and rebalancing the activity of NK cells to re-establish a healthy ratio of CD4^+^/CD8^+^ T cells [226].

Similarly, squalene, a chemical component of extra virgin olive oil, has been implicated in modulating the innate immune system’s response and was demonstrated to regulate inflammatory and wound-healing responses [227]. The effects of squalene have also been investigated in regulating the adaptive immune response [228]. In this report, a TLR-4 glucopyranosyl lipid adjuvant was prepared and administered in an emulsion of squalene oil and water and promoted a robust CD4^+^ response.

Furthermore, apigenin, a flavonoid compound found in green fruits and vegetables such as celery and parsley, was demonstrated to have immune-modulatory and anti-tumor activity when administered in conjunction with PD-1/PD-L1 checkpoint blockade [229]. Specifically, in xenograft models of melanoma, increased T cell infiltration was observed in groups treated with apigenin in addition to reducing tumor growth. Collectively, these naturally occurring compounds provide a rationale for the inclusion of dietary supplements in an immunotherapeutic treatment regimen to promote a robust, synergistic immune response that may improve patient response rates.

### 4.2. Re-Educating the Gut Microbiota to Modulate the Immune Response

The crucial role of the gut microbiota, both in composition and function [228,229,230], has become increasingly evident in human physiology. The intricate balance and homeostasis of the gut microbiota are necessary for normal human physiology, with any perturbations associated with many chronic diseases such as cancer, especially colorectal carcinoma [230]. This is due to gut microbiota’s ability to act as an endocrine organ and stimulate an immune response, regulating inflammatory or metabolic diseases and infectious diseases [231,232]. Microbiota can impact cancer development via chronic inflammation maintenance or directly influence immune cells [233]. Thus, it is not surprising that consuming a diet that nurtures a healthy gut microbiota is crucial for overall human health, as several micronutrients and fibers play a critical role [234].

Interestingly, diet supplementation with arginine, omega-3 fatty acids, and nucleotides has been shown to significantly improve the immune response of cancer patients undergoing surgery and to reduce infectious complications, duration of hospital stay, and comorbidities [235]. Additionally, polyphenols, found in vegetables, fruits, cereals, extra virgin olive oil, tea, and coffee, demonstrate antioxidant and anti-inflammatory properties and have been evaluated for their anti-cancer efficacy. For example, polyphenols have inhibited breast cancer cell proliferation and metastasis and downregulated IL-6 production in vitro and in vivo [236]. Furthermore, increasing evidence that clinical benefits, derived from phytochemicals, can be achieved from concentrations that exceed the maximum achieved via dietary consumption, have demonstrated promise in modulating the immune system [237]. For example, shikonin, a plant metabolite, has been reported to induce an immune response from dendritic cells to decrease tumor growth and metastasis [238]. As a result, efforts are underway to produce these and other phytochemical compounds’ derivatives to harness their potential to modulate and enhance immunotherapies’ effectiveness. Mushrooms have also been associated with stimulating the immune response, modulating humoral and cellular immunity, and improving antimutagenic and antitumorigenic activity. One of the primary mechanisms by which mushrooms modulate the immune system to fight tumors is through the polysaccharide β-glucans in their cell walls, associated with the increased phagocytic activity, cytokine production, and reactive oxygen intermediates [239]. Due to these effects, metabolites derived from mushrooms are promising immunomodulation agents for cancer immunotherapy, particularly for individuals with an immune system weakened by radiotherapy and chemotherapy [239].

Bioactive compounds from dietary food or derived from food have health benefits in addition to fundamental nutritional value [223]. Thus, modulating dietary intake can be used to improve current therapeutic options. Combining nutrition with current or emerging immunotherapies highlights a new avenue of cancer therapy; however, additional research is necessary to optimize techniques to harness the maximal benefits from the nutritional factors that modulate the immune response.

## 5. Discussion

When considering conventionally administered treatments for many challenging cancers, such as breast, pancreatic, lung cancer, and neuroblastoma, the emerging challenge is to upend their highly adaptive disease progression. Cancers have many compensatory pathways that can be engaged to overcome the therapeutic effects of conventional treatments. Many of these compensatory pathways include activating survival pathways that can lead to an aggressive malignancy that is more challenging to treat during a follow-up round of treatment. As a result, a combination approach to target those pathways that account for cancer’s compensatory nature is mandatory. In this review, we outline a similar approach to improving immunotherapy effectiveness by supplementing immunotherapies with a combination of immunomodulating agents that support the immune system and enhance the therapeutic outcomes of immunotherapy. Here, an improved clinical response could be observed. As a study model, we provide a brief overview of how a combination approach could be applied to treat NB.

Several complex immunotherapeutic options to treat patients with high-risk NB are currently the standard of care; however, several limitations are associated with this treatment approach that may be reduced by applying the proposed alternatives presented in this review. The post-consolidation standard of care therapies incorporates a combination of dinutuximab (a monoclonal antibody that targets the disialoganglioside (GD-2) (a glycolipid expressed on NB cells), cytokines (IL-2 and GM-CSF), and isotretinoin (a retinoid agent that induces differentiation of immature NB cells) [240]. Side effects of this treatment regimen include capillary leak and hypotension, hypersensitivity to the toxicities associated with IL-2 or dinutuximab, and neuropathic pain [240]. One potential option to overcome these side effects is to offer a targeted approach rather than systemic delivery. For example, a combination of CAR T cell and oncolytic therapy could permit personalized local treatment that may overcome many of the toxicities associated with systemic administration of the therapeutics. As detailed, CAR T cell therapy offers a personalized and targeted method that would essentially allow T cells to target specific receptors located on NB cells [241]. NB’s two common targets include the GD-2 and *MYCN* gene, both uniquely expressed on NB [242]. CAR T cells targeting GD-2 and the *MYCN* gene are currently in the early phases of clinical trials. It has primarily established the safety and efficacy of this treatment option [241]. However, many of the challenges associated with this approach include T cell exhaustion and an immunosuppressive tumor microenvironment [241]. Therefore, supplementing this regimen with oncolytic viral therapy is one method to enhance CAR T cell therapy’s effectiveness.

The Zika virus presents a unique vector that has demonstrated preclinical success in NB mouse models, given the virus’s ability to cross the blood-brain barrier [159]. One method to improve these preliminary studies would be developing patient-derived xenografts (PDX) models by obtaining primary NB tumors from high-risk NB patients to study the efficacy of the Zika virus approach. Additionally, preliminary reports have shown that this virotherapy can target neural CSCs, eliminating the need for isotretinoin and overcoming the toxicities associated with this agent [240]. However, as detailed, the Zika virus has several drawbacks, including infecting neural cells. Therefore, an alternative option could be BCG’s application to target the hypoxic conditions in NB and the cancer stem cells typical of this niche.

Additionally, preliminary data have shown that the Zika virus preferentially targets CSCs, as evidenced by an increase in SOX-2 cancer stem cells infected by the Zika virus [159]; however, there is a clear correlation between the Zika virus and hypoxic regions of NB tumors that were not established in that study. In contrast, BCG’s application has been shown to target CSCs in the hypoxic niche [173] and may potentially overcome the limitations associated with the Zika virus. Promising results with the application of BCG have shown that a robust immune response is possible. Although this is an application of BCG is in a state of infancy, this treatment approach may have significant implications on treating NB. In-depth experimental explorations will be required to assess the efficacy of this approach.

Finally, the dual immunotherapy method presented, using a CpG vaccine coupled with anti-OX-40 therapy, is a potential approach to treating NB. However, one primary caveat needs to be addressed. It is essential to establish whether NB tumors are infiltrated by Treg immune cells. As detailed, Tregs have been implicated in promoting an immune-suppressive TME and supporting tumor growth. In patients presenting with NB, an increase in Tregs’ systemic circulation has been reported [243]; however, it has yet to be determined whether Tregs are present in NB tumors. In a pre-clinical animal study currently underway by our group, we found that the depletion of Tregs impacted the growth of NB tumors. These data indicate the critical function Tregs may have in the progression of NB tumors.

Further investigations into characterizing the presence of Tregs in NB tumors, using NB mouse models, would offer insights into whether dual immunotherapy would be beneficial. Additional methods to explore this dual immunotherapy treatment’s efficacy would be to develop an in vitro human NB model [244,245,246]. For example, a model could incorporate a three-dimensional (3D) co-culture system to recapitulate the unique conditions of the TME while also including Tregs to assess how the in vitro NB spheroid develops and whether dual therapy is feasible [247,248]. Previous studies have been done on colorectal cancer cell spheroids and breast cancer spheroids with T and NK cells, providing a viable platform for studying tumor-lymphocyte interactions’ antitumor potential for immunotherapy [249,250]. However, detailed characterization studies would need to be completed to assess the Tregs and ligand expression in human NB samples to generate a 3D co-culture system.

Although more advanced immunotherapeutic approaches to treat NB are currently in the early stages, the promising results and applications presented in this review offer exciting new options to explore.

## 6. Conclusions

Immunotherapy was voted the breakthrough advance of the year by Science in 2013 because of the responsiveness observed in patients presenting various cancers. Although this therapeutic approach has promising implications in the clinic, this treatment’s relative responsiveness has limitations that thus far have reduced its effectiveness to use in small subsets of patients within a larger cohort, as presented in Figure 1. This review offers several methods to enhance this promising treatment approach’s effectiveness, as depicted in Figure 2. Firstly, it is crucial to consider the detrimental effects of cancer stem cells in driving the patients’ reduced responsiveness. We propose immunotherapy using CAR T cell therapies to target cancer stem cell biomarkers to solve this challenge. Secondly, we propose combining virotherapy and immune checkpoint inhibitors to induce an immunogenic TME to improve the outcome of immune checkpoint inhibitors. Additionally, the emerging approaches incorporating genetic and epigenetic therapies combined with immunotherapeutic strategies need to be explored. We also provide a brief discussion on the implications of the ongoing pandemic and the challenges presented by COVID-19 for cancer patients. Thirdly, we propose the means to incorporate dietary and nutrigenomic approaches to enhance and support the immune system, combined with immunotherapy to elicit a robust immune response against the tumor. Fourthly, considering the multi-mutagenic profiles of cancer, targeting metabolic syndrome could be considered an important therapeutic strategy for solid and non-solid tumors. The concept of metabolic syndrome is one of the critical survival mechanisms of cancer cells. Finally, we propose a model to test the efficacy of the new immunotherapy approaches in a neuroblastoma model, as schematically presented in Figure 3. Combining these solutions could have positive implications in the clinical outcomes observed following immunotherapy and should be investigated.

## Figures and Tables

**Figure 1 cancers-13-03596-f001:**
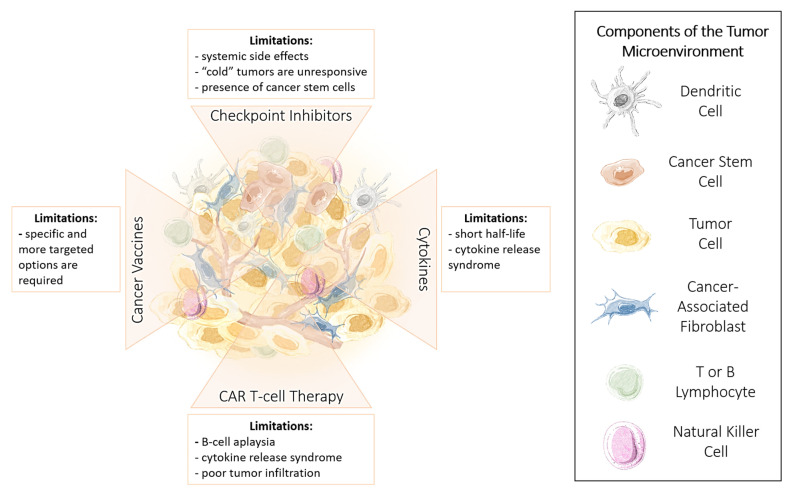
Limitations of Immunotherapy. There are four main categories of immunotherapy: immune checkpoint inhibitors (i.e., PD-1/PD-L1 and CTLA-4), cytokines (i.e., IFN-α), cancer vaccines, and chimeric antigen receptor T cells (CAR T cell therapy). Although these therapeutic options have been successful in the clinic, only a small subset of patients respond. Several limitations of this approach are summarized in the schematic. Abbreviations: CTLA-4, cytotoxic T-lymphocyte-associated protein 4; IFN-α, interferon-alpha; PD, programmed cell death protein-1.

**Figure 2 cancers-13-03596-f002:**
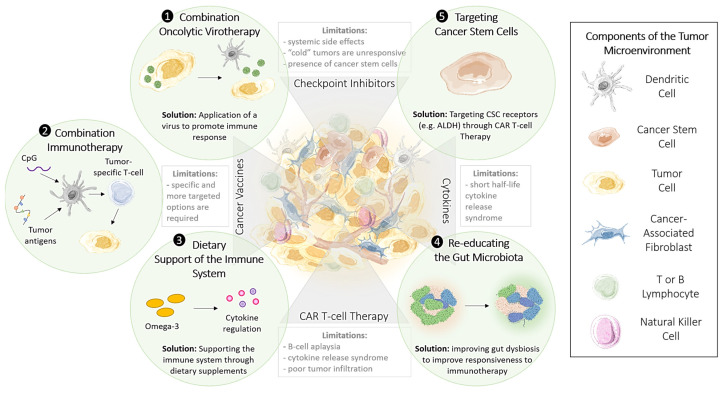
Solutions to overcoming the limitations of immunotherapy. Several viable options could potentially overcome the limitations presented by immunotherapies. (**1**) Targeting cancer stem cells; given their unique biology, are not currently targeted by conventional immunotherapies. (**2**) A combination therapeutic approach, using oncolytic viruses to promote a more robust immune response upon tumor cell lysis, would release tumor antigens into the microenvironment. These tumor-specific antigens would then be recognized by immune cells and could elicit a robust response and enhance tumor cells’ clearance. (**3**) Combination immunotherapy using a CpG vaccine to prime dendritic cells to recognize antigens, followed by T cells’ presentation, could lead to a more localized and targeted systemic anti-tumor immune response. (**4**) Dietary supplements to support the immune system, and finally, (**5**) re-educating the gut microbiota may improve the immune system’s competency and improve patient response to immunotherapy. Abbreviations: ALDH, aldehyde dehydrogenase; CAR T cells, chimeric antigen receptor T cells.

**Figure 3 cancers-13-03596-f003:**
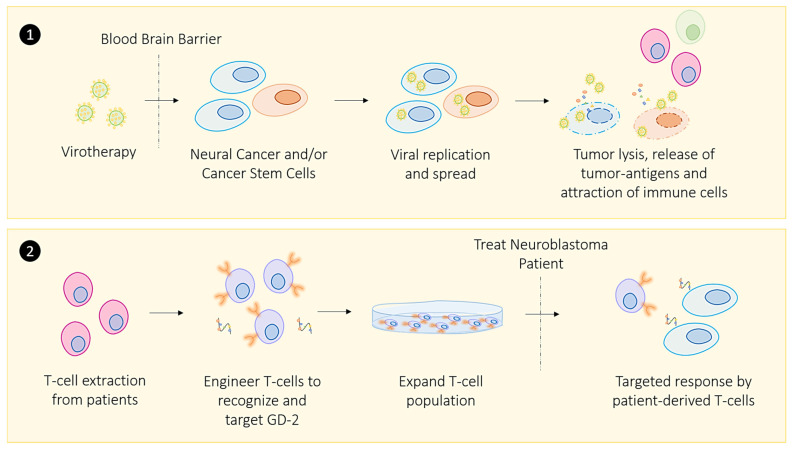
Neuroblastoma as a Study Model. Schematic representation of the virotherapy and targeted T cell therapeutic approach is applied to treat neuroblastoma. (**1**) Application of Zika virotherapy is engineered to target neural cancer and cancer stem cells to elicit an immune response targeting tumor-associated antigens’ release following virus-mediated cancer cell lysis. (**2**) Targeted CAR T cell therapy, where patient-derived T cells are engineered to recognize and target antigens exclusively expressed on neural cancer cells (i.e., GD-2) and target and clear cancer cells expressing this marker. Abbreviations: CAR T cells, chimeric antigen receptor T cells; GD-2, disialoganglioside.

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
