# Peer review of "The Next-Generation of Combination Cancer Immunotherapy: Epigenetic Immunomodulators Transmogrify Immune Training to Enhance Immunotherapy"

_cancers, 2021, doi:10.3390/cancers13143596_

Round 1
Reviewer 1 Report
The paper by Dr. Bayat Mokhtari and colleagues is a thorough review on what is going on with new combination therapies to increase immunotherapy effectiveness.
The review is clearly written, and the contents carefully explain the main checkpoint inhibitors used at this moment as anti-cancer treatment along with the main reasons for its failure. It is obvious that although immunotherapy results have been promising, its effectiveness highly varies with cancer type.
So, combination therapies seem to be the next logical step in cancer treatment to circumvent the development of resistance mechanisms. Following this lead, authors offer up to four strategies to enhance immunotherapy usefulness in the future.
I think the study would benefit from adding a brief explanation on hyperprogressive disease described with immunotherapy and by completing with the already known role of physical exercise in boosting immune system.
Authors should correct several typo errors in the text.
Author Response
Author response: Thank you for the additional comments to improve the manuscript. We have incorporated a brief explanation on hyperprogressive disease (HPD) described with immunotherapy, and the potential underlying mechanism(s) behind HPD with the inhibition of PD-1 lies in the tumor microenvironment (TME) (Lau and Leighl [18] Journal of Thoracic Disease 2019, S1877-S1880). We \have also included the reports by Angelicola, S.; Ruzzi, F.; Landuzzi, L.; Scalambra, L.; Gelsomino, F.; Ardizzoni, A.; Nanni, P.; Lollini, P.-L.; Palladini, A. IFN-γ and CD38 in Hyperprogressive Cancer Development. Cancers 2021, 13, 309, and Camelliti, S.; Le Noci, V.; Bianchi, F.; Moscheni, C.; Arnaboldi, F.; Gagliano, N.; Balsari, A.; Garassino, M.C.; Tagliabue, E.; Sfondrini, L., et al. Mechanisms of hyperprogressive disease after immune checkpoint inhibitor therapy: what we (don’t) know. Journal of Experimental & Clinical Cancer Research 2020, 39, 236, doi:10.1186/s13046-020-01721-9.
We have also include a section on role of physical exercise in boosting immune system.
3.3 Exercise-Induced Epigenetic Modifications in Genes
Regular physical activity can help to improve overall health. Increasing evidence supports the premise that exercise can result in, or can be mediated by, epigenetic modifications [198]. Environmental factors, including diet [199] and physical exercise [198], have been reported to alter the epigenome. Barrón-Cabrera et al. [200] reported an excellent systemic review on the epigenetic modifications resulting from exercise interventions. Interestingly, the report describes exercise-induced modifications in genes related to (a) insulin resistance and type-2 diabetes, (b) obesity, (c) inflammation, and (d) cardiovascular disease and blood lipid alterations. These studies indicate that exercise interventions can alter the epigenome and that these outcomes could be related to specific metabolic pathways. However, there are several limitations in the studies reported to date, including small sample size, heterogeneous populations, the different exercise interventions, the exercise tests available, and the different epigenetic modifications measured in different tissues.
Additional writing on this section is included.

Reviewer 2 Report
The authors must include more details about therapies related to CAR-T cells and whats the major reasons that it failed in the clinic as well as how studies are underway to improve on that aspect.
Author Response
Author response: Thank you for the additional comments to improve the manuscript. We have included additional details about the therapies related to CAR T cells. For example, the limitations and challenges relating to CAR T cells include severe toxicities, restricted trafficking, infiltration into and activation within tumors, suboptimal persistence in vivo, antigen escape and heterogeneity, and manufacturing issues [79,80]. While many CAR T cell therapy patients are in remission and have prolonged survival, the long-term effects remain unclear and are under current investigation [81]. CAR T cells can cause cytokine release syndrome (CRS), CAR-T cell-related encephalopathy syndrome (CRES) and neurotoxicity [82-85]. For example, CRS is common with patients undergoing CAR T therapy in developing multiorgan failure. Also, neurotoxicity/CRES can result in seizures and cerebral edema. Graham et al. [84] propose centers be JACIE (Joint Accreditation Committee-ISCT & EBMT) with accredited bone marrow transplant (BMT) units, alongside hematologists, intensivists, neurologists, cardiologists and renal specialists.
Under some circumstances, particularly in solid tumors with distinct microenvironments, the injected CAR T cells do not persist, limiting therapy efficacy. However, one of the significant challenges in CAR T cell therapy is developing CAR T cells' complex and costly process [86]. Rafiq et al. [79] addressed these issues with crucial points addressing the current CAR T obstacles with a wide range of engineering strategies. For example, optimal molecular design of the CAR T is achievable through examining the many variations of the constituent protein domains. Secondly, the current toxicities associated with CAR T cell therapy can be mitigated using engineering strategies to make CAR T cells safer by overcoming on-target, off-tumor toxicities. Thirdly, the efficacy of CAR T therapy can be enhanced with engineering strategies addressing the various challenges relating to haematological and solid malignancies.
In preclinical studies of hematological cancers, Barros et al. [87] develop a mathematical platform to enable in silico experiments to investigate the interplay between tumor cells, effector, and memory CAR-T cells in immunodeficient mouse models. They found that CAR T therapy effectiveness mostly depends on the differentiation of effector to memory CAR T cells, CAR T cytotoxic capacity, tumor growth rate, and tumor-induced immunosuppression. Interestingly, using the HDLM-2 tumor model with low tumor proliferation and a less aggressive tumor, the CAR T cell therapy was effective on tumor elimination and enhancing memory CAR T cells. The model captured tumor elimination after CAR T immunotherapy with new tumor challenges due to memory CAR T cells’ long-term protection. In contrast, the RAJI model, with its high proliferation and escape from CAR T cell immunotherapy, identified the effect of indoleamine-2,3-dioxygenase 1 enzyme (IDO) overly expression by the RAJI cells, impacting CAR T cell immunotherapy and their combination with an IDO inhibitor. Importantly, it was noted that the CAR T cell dose determination for a given tumor burden is a critical factor in immunotherapy success [87].

Reviewer 3 Report
In the review manuscript, Reza Bayat Mokhtari et al. describe cancer immunotherapy approaches focusing on a combination of the different types, including epigenetic regulation and trained immunity, in order to increase the effectiveness. The topic is of great interest. There are high hopes for combination immunotherapy to overcome the problem of variability in the patient’s response to the treatment. The review is very informative, authors collected a large amount of data related to immunotherapeutic approaches to treat cancer. They consider immune checkpoint inhibitors, cytokines, CAR-T cells, oncolytic viruses, CpG nucleotides, epigenetic reprogramming, agonists of activating receptors, targeting cancer stem cells, dietary supplementation and some other therapy types that can be used in combinations for cancer patient treatment. In the Discussion section the authors describe a model of combined immunotherapy of neuroblastoma. The manuscript is well-structured, carefully written, and beautifully illustrated. I would recommend this manuscript for publication in Cancers.
Still, several minor concerns may be addressed to improve the overall quality of the manuscript.
- In general, the manuscript looks unnecessarily wordy and repetitive in some parts. I would pretend the authors reread the text in order to dispose of word or semantic repetitions.
- Some of the abbreviations are introduced twice or in not proper place (for example: EMT – lines 388, 417; ALDH – lines 381, 440).
- The term “T-cells” is commonly used as “T cells” without hyphen.
- There are several awkward phrases (for example, lines 577, 673)
- The title of the section 1.4 should be changed to “Immune checkpoint inhibitors”.
- Line 647: should be “intratumoral” instead “intertumoral”.
Author Response
In general, the manuscript looks unnecessarily wordy and repetitive in some parts. I would pretend the authors reread the text in order to dispose of word or semantic repetitions.
- Some of the abbreviations are introduced twice or in not proper place (for example: EMT – lines 388, 417; ALDH – lines 381, 440).
- The term “T-cells” is commonly used as “T cells” without hyphen.
- There are several awkward phrases (for example, lines 577, 673)
- The title of the section 1.4 should be changed to “Immune checkpoint inhibitors”.
- Line 647: should be “intratumoral” instead “intertumoral”.
Submission Date14 June 2021Date of this review10 Jul 2021 02:20:59
Author response: Thank you for the additional comments to improve the manuscript. Items listed 1-5 have been addressed in the revised manuscript. We also highlighted repetitive sections to be cross-out.
